# Disentangled Skill Representations for Predictive Human Modeling

## Abstract

Understanding human skill is essential for AI systems that collaborate with, coach, or assist people. Unlike typical latent variable estimation problems—which rely on single observations or explicit labels—skill is a persistent, compositional, and behaviorally grounded construct that must be inferred from patterns over time. We introduce Skill Abstraction with Interpretable Latents (**SAIL**), a method for learning disentangled skill representations from naturalistic behavioral data. Our approach produces a skill embedding that is robust to spurious performance fluctuations and captures core, transferable representation of human subskills. Furthermore, **SAIL** supports skill-informed behavior prediction that generalizes across a variety of contexts. We represent each individual with a persistent skill embedding that controls a blend between expert and novice behavior bases and is trained using counterfactual subskill swaps for disentanglement. This design yields a representation that is both robust to performance variation and structured for interpretability. We demonstrate that **SAIL** outperforms prior methods across two domains—high-performance driving and baseball batting—producing skill representations that are stable, predictive, and interpretable.

## 1 Introduction

AI systems that support, collaborate with, or coach humans must infer skill from behavior to personalize instruction, coordinate effectively, and adapt assistance to user ability. Yet skill is difficult to model: it is latent, temporally extended, and behaviorally grounded, requiring inference from patterns across multiple trials rather than individual outcomes. Moreover, skill has a compositional structure, reflecting subskills that improve at different rates or shape distinct aspects of behavior. These properties distinguish human skill from other variables in representation learning (e.g., object identity in computer vision or task-level skills in robotics) and make its assessment critical for personalization.

Unlike in robotics, where "skill" often refers to reusable action primitives or options (Botteghi et al., 2025; Lesort et al., 2018), we focus on *human skill*, a persistent, individual-level construct composed of distinct subskills that evolve at different rates and shape different aspects of behavior (Newell, 1991; Ericsson et al., 1993). Unlike performance, which can vary trial to trial, skill must be inferred from patterns across time, making its stable representation particularly challenging. [1]

Our desiderata for skill modeling as a representation learning problem are as follows: (1) **Construct Validity (Messick, 1995)**: The representation should capture human skill while remaining robust to noise and style, varying across individuals but stable across sessions and contexts for the same person. (2) **Predictive Utility**: The learned representations should have good predictive power and support forecasting of behavior observations and outcomes. (3) **Interpretability**: The representations should yield disentangled subcomponents corresponding to distinct subskills that can be easily identified by human experts.

To satisfy these desiderata, we introduce **S**kill **A**bstraction with **I**nterpretable **L**atents or **SAIL**. **SAIL** is explicitly designed to promote construct validity, generalization, and interpretability in the learned

---

[1] In psychology and motor-learning theory, performance is considered a transient expression of skill influenced by situational factors (Iso-Ahola, 2024; Fitts & Posner, 1967) We adopt this usage throughout.

skill representation. **SAIL** learns a persistent individual-level skill embedding. This design aggregates information across multiple behavioral observations from an individual, allowing the model to abstract away trial-level noise and capture consistent, long-term behavioral tendencies that reflect true skill yielding a stable learned representation. While simple aggregation methods (e.g., averages or Elo scores) can smooth variability, they conflate skillful risk-taking with poor performance. For example, a spinout in racing may raise average lap time even though it reflects an expert pushing the vehicle to its limits. Our approach learns a structured skill representation that contextualizes such events—abstracting away trial-level noise while preserving the subskill structure necessary to explain and predict behavior.

Rather than decoding behavior directly from the learned skill embedding, **SAIL** predicts behavior as a blend of canonical novice and expert behaviors. The intuition behind the blending formalism is that it removes low-level behavioral variability and forces the representation to encode structured, high-level skill-relevant variation. Overall, the model learns not only to capture general ability via the skill embedding but also how it maps onto behavior in different contexts. To encourage interpretability and disentanglement, we supervise the skill space with behaviorally grounded subskill metrics and introduce a counterfactual training strategy. This counterfactual procedure ensures that each latent dimension governs a distinct and identifiable aspect of skill.

We evaluate **SAIL** in both a high-performance driving domain and a baseball batting domain. Sports provide a natural testbed for skill modeling: success depends on mastering multiple subskills, and expertise is expressed through consistent, structured patterns of behavior. These domains also offer rich, multimodal data that capture both outcomes and the processes that generate them, and success can be quantified in well-defined terms such as lap time, ranking, or batting performance.

In this work we contribute the following:

1. Formulate human skill modeling as a representation learning problem, with explicit desiderata of construct validity, predictive utility, and subskill interpretability.
2. Propose **SAIL**, a method that combines person-level embeddings, expert–novice basis blending, and counterfactual subskill swapping to induce a disentangled and predictive skill space.
3. Demonstrate effectiveness across two distinct domains—high-performance driving and baseball— and show that **SAIL** outperforms baselines in producing skill representations that are stable, generalizable, and interpretable.

## 2 RELATED WORK

Modeling human skill is a long-standing challenge across education, sports science, robotics, and human–AI interaction Anderson (2014); Ericsson et al. (1993). Skill is a latent construct that can be inferred from behavior over time Newell (1991); Schmidt et al. (2018). Traditional metrics for capturing ability such as completion time, accuracy, or error rates Fitts & Posner (1967) are noisy and context-dependent and tend to capture *performance* rather than true skill. Psychometric methods like Item Response Theory and Bayesian Knowledge Tracing Embretson & Reise (2013); Corbett & Anderson (1994); Piech et al. (2015) provide principled ability estimates but remain tied to discrete items.

More advanced approaches such as trajectory clustering and inverse reinforcement learning (IRL) Ziebart et al. (2008); Abbeel & Ng (2004) aim to uncover latent behavioral structure, and extensions using probabilistic embeddings introduce latent variables $z$ that can be interpreted as skill. However, these methods do not yield interpretable subskill structure or the granularity required for continuous motor behavior Guadagnoli & Lee (2004); Wulf (2016). Similarly, work in robot teaching emphasizes the importance of explicitly decomposing tasks into skills and subskills (Argall et al., 2009; Cakmak & Thomaz, 2012).

**Representation Learning for Human Behavior:** Building on general insights from representation learning Bengio et al. (2013), researchers have sought to compress sequences of states and actions into embeddings that summarize behavior Zhang et al. (2019); van den Oord et al. (2018). Autoencoders and sequence VAEs provide compressive frameworks Kingma & Welling (2013), while contrastive and self-supervised methods (e.g., CPC, SimCLR) learn robust embeddings (van den Oord et al., 2018; Chen et al., 2020b). However, these approaches operate on per-instance data,

often reflecting context or style rather than stable skill (Varona-Moya et al., 2021; Wu et al., 2021; Sun et al., 2024).

Recent work on representation learning tailored for humans and human interaction has focused on capturing behavioral regularities relevant for collaboration, including adaptation to novel partners Jacques et al. (2019), optimizing shared autonomy in assistive settings Gopinath et al. (2017), predictive world models of human intent for shared control DeCastro et al. (2024), and low-dimensional manifolds for intuitive shared autonomy Jeon et al. (2020). A common theme in this literature is the need to aggregate across trajectories and to use predictive objectives that capture semantic structure more effectively than reconstruction, as argued in JEPA (LeCun, 2022). Our approach builds on these insights by requiring that skill embeddings demonstrate *predictive validity*: they should anticipate future behavior and downstream performance. To encourage this property, we introduce expert–novice blending as a structured inductive bias on how skill trajectories evolve.

**Disentangled and Interpretable Representations:** Disentanglement seeks latent dimensions that map onto distinct, interpretable factors. Methods like Beta-VAE, InfoGAN, and FactorVAE Higgins et al. (2017); Chen et al. (2016); Kim & Mnih (2018) encourage structured representations but face trade-offs in fidelity, independence, and identifiability Locatello et al. (2019). For skill modeling, this limits alignment between latent variables and subskills (Zhang et al., 2021). Recent methods such as DUSDi (Hu et al., 2024) aim to decompose skills into interpretable components affecting distinct environment factors, but challenges remain.

**Modeling Skill in Robotics:** Several works in reinforcement learning and robotics have explored learning latent skill spaces for control policies. Hausman et al. (2018) and Petangoda et al. (2019) learn disentangled or transferable skill embeddings to enable policy reuse and compositional action generation across tasks, while Gupta et al. (2018) develop meta-reinforcement learning strategies that encourage structured exploration. More recently, Dave & Rueckert (2025) propose a kernel-based approach for skill disentanglement within continuous control. However, these approaches address *robotic control skill* - that is, the discovery of reusable action primitives or policies for task execution - rather than *human skill* as a persistent, compositional, and interpretable construct. Our work differs fundamentally in scope and objective: SAIL seeks to model how human skill manifests, evolves, and generalizes across contexts, focusing on cognitive and behavioral interpretability rather than motor control or policy transfer.

Our method unifies advances in representation learning, disentanglement, and counterfactual reasoning. By introducing participant-specific embeddings, novice–expert basis blending, and counterfactual subskill swaps, we produce skill representations that are stable, predictive, and interpretable.

## 3 APPROACH

**Problem Formulation:** We aim to learn a latent representation of human *skill* from behavioral data. Let $\mathcal{D} = \{\tau_1, \ldots, \tau_N\}$ be a set of trajectories, where each $\tau_i = \{(s_i^t, a_i^t)\}_{t=1}^{T_i}$ is a sequence of states and actions with $T$ timesteps and $D$ trajectory features performed in a task context $c \in \mathcal{C}$ (e.g., a racetrack or batting condition). Trajectories may include multimodal features such as vehicle telemetry, gaze, or body kinematics.

Our goal is to infer $z_s \in \mathbb{R}^d$ that captures stable, individual-level skill across trajectories and contexts. We distinguish *skill*, a persistent construct, from *performance* (Iso-Ahola, 2024), which reflects trial outcomes (e.g., lap time) and is sensitive to environmental variability. Skill should be invariant to context $c$ while remaining *compositional*, with $z_s$ decomposing into interpretable subcomponents $z_s^{(k)}$ corresponding to distinct subskills (e.g., vehicle handling, gaze) Newell (1991); Anderson (1982). This structure enables targeted probing of specific deficiencies.

To connect latent subskills with behavior, we use *skill metrics* $m \in \mathcal{M}$: measurable quantities derived from trajectories, expert annotations, or auxiliary tasks that act as noisy proxies. For example, peak lateral g-force in a skidpad drill serves as a proxy for vehicle handling Schrum et al. (2025). Multiple metrics may map to the same subskill, providing supervision for learning structured representations of $z_s$. Our approach is detailed in Alg. 1

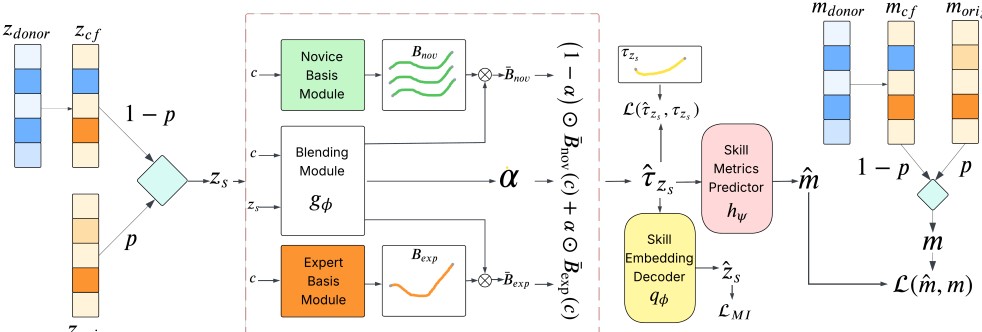

Figure 1: **SAIL** overview: Each participant is associated with a persistent skill embedding $z_s$ that aggregates behavior across trials. This embedding controls a blend between expert and novice basis behaviors to predict trajectories, abstracting away trial-specific noise. The embedding is partitioned into subskill slices, which are supervised with behaviorally grounded skill metrics and trained using counterfactual swaps to encourage disentanglement and interpretability.

### 3.1 PARTICIPANT-SPECIFIC SKILL EMBEDDING

A naïve approach to skill representation would be to encode each behavioral trajectory into a latent space. However, single trajectories often reflect transient influences such as noise, fatigue, or environmental variation, making representation learning from single trajectories more suitable for capturing performance as opposed to stable skill characteristics. Moreover, isolated trajectories contain no inherent link to the individual who produced them, which is essential if skill is to be modeled as a persistent, person-specific construct. To address these issues, **SAIL** assigns each participant a persistent skill vector $z_s \in \mathbb{R}^d$, learned jointly with the model parameters (Alg. 1 Line 1). This design pools information across multiple trajectories from a participant, akin to how user and speaker embeddings are learned for recommendation(Koren et al., 2009) and speech recognition systems (Snyder et al., 2018), abstracting away trial-level variability and capturing the long-term participant-specific behavioral tendencies that define skill.

$z_s$ is treated as a subject-specific learnable parameter that is refined throughout training based on behavioral evidence. At test time, the model is frozen and the behavioral trajectories loss is used to embed the test subjects. Conceptually, $z_s$ is an explanatory variable: skill generates behavior, not the other way around. To prevent trivial collapse, we introduce an auxiliary network $q_\phi$ that reconstructs $z_s$ from generated trajectories. This provides a variational lower bound on the mutual information between $z_s$ and predicted behavior, encouraging the embedding to encode information that is both behaviorally meaningful and recoverable from observed trajectories (Kingma & Welling, 2013; Chen et al., 2016): $\mathcal{L}_{\text{MI}} = -\mathbb{E}_{\tau \sim p(\tau|z_s)}[\log q_\phi(z_s|\tau)]$.

### 3.2 NOVICE-EXPERT BASIS BLENDING

Our goal is to represent skill in a way that abstracts away trial-specific noise and stylistic variation while preserving stable, skill-relevant structure. We assume that observed behavior is generated from the skill embedding and lies on a spectrum that ranges from novice to expert performance. To operationalize this, we introduce a *novice–expert basis* that provides canonical reference behaviors against which individual skill can be expressed. This design rests on the assumption that skill varies in a continuous and interpolatable manner—i.e., that intermediate behaviors can be meaningfully represented as blends between novice and expert bases. While this continuity is an abstraction, it captures the intuition that progression in skill is gradual and structured, and allows our model to interpolate skill levels and generalize across individuals.

For each task context $c$ (e.g., a racetrack in driving or batting condition in baseball), we define a set of expert and novice bases trajectories:

$$B_{\text{exp}}(c) = \{B_{\text{exp}}^{(1)}(c), \ldots, B_{\text{exp}}^{(M)}(c)\}, \quad B_{\text{nov}}(c) = \{B_{\text{nov}}^{(1)}(c), \ldots, B_{\text{nov}}^{(K)}(c)\},$$

---

**Algorithm 1** SAIL: Skill Abstraction with Interpretable Latents

---

**Require:** Trajectories $\tau_i$, contexts $c_i$, subskill metrics $m_i$

1: Initialize participant embeddings $z_{s,i} \sim \mathcal{N}(0, 0.1)$
2: **for** each training iteration **do**
3:     Sample batch of participants and trajectories
4:     Predict behavior $\hat{\tau}_{z_s}$ via expert–novice blending (Sec. 3.2)
5:     Decode predicted subskill metrics $\hat{m} = h_\psi(\hat{\tau}_{z_s})$
6:     Compute total loss: $\mathcal{L} = \lambda_{\text{recon}}\mathcal{L}_{\text{traj}} + \lambda_{\text{metric}}\mathcal{L}_{\text{subskill}} + \lambda_{\text{MI}}\mathcal{L}_{\text{MI}}$.
7:     **if** counterfactual step (probability $1 - p$) **then**
8:         Swap subskill slice $z_{\text{orig}}^{(k)} \leftarrow z_{\text{donor}}^{(k)}$ and metric $m_{\text{orig}}^{(k)} \leftarrow m_{\text{donor}}^{(k)}$
9:         Reconstruct counterfactual trajectory $\tilde{\tau}_{\text{orig}}$ from $\tilde{z}_{\text{orig}}$
10:        **Skip trajectory reconstruction loss; apply metric loss only for swapped subskill** $k$
11:     **end if**
12:     Update model parameters and participant embeddings jointly via back-propagation
13: **end for**

---

where each basis trajectory $B^{(i)} \in \mathbb{R}^{T \times D}$ represents a canonical behavior pattern drawn from the extremes of the skill distribution, with $T$ timesteps and $D$ trajectory features. The expert set could in principle capture multiple distinct high-skill strategies, while the novice set spans the heterogeneous modes of novice performance (e.g., over-cautious, inconsistent, or poorly timed execution). In practice we find that a single expert basis is sufficient as expert demonstrations tend to cluster tightly around a consistent solution. In contrast, we retain multiple novice bases to capture the diversity of novice behavior.

These bases can be derived in multiple ways: directly from demonstration data, learned jointly with the embedding, or generated by an optimal controller. In practice, we find that a simple yet effective construction works well: (1) use trajectories from the most expert demonstrator to define $B_{\text{exp}}$, and (2) apply principal component analysis (PCA) to a collection of novice trajectories to define $B_{\text{nov}}$, capturing the dominant axes of novice variability. This design allows the model to interpret $z_s$ in terms of blending toward or away from the expert solution along meaningful novice dimensions. The skill embedding is then mapped into blending coefficients for each basis through a blending module network $g_\phi$:

$$\alpha = g_\phi(z_s, c) \in [0, 1]^{M \times K \times T \times D}.$$

Finally, the predicted behavior is expressed as a weighted combination over the bases where the weights are learned via $g_\phi$ (Fig 1):

$$\bar{B}_{\text{exp}}(z_s, c) = \sum_{m=1}^{M} w_{\text{exp}}^{(m)}(z_s, c)\, B_{\text{exp}}^{(m)}(c), \quad \sum_{m=1}^{M} w_{\text{exp}}^{(m)}(z_s, c) = 1, \; w_{\text{exp}}^{(m)} \geq 0$$

$$\bar{B}_{\text{nov}}(z_s, c) = \sum_{k=1}^{K} w_{\text{nov}}^{(k)}(z_s, c)\, B_{\text{nov}}^{(k)}(c), \quad \sum_{k=1}^{K} w_{\text{nov}}^{(k)}(z_s, c) = 1, \; w_{\text{nov}}^{(k)} \geq 0$$

$$\hat{\tau}_{z_s} = \alpha \odot \bar{B}_{\text{exp}}(z_s, c) + (1 - \alpha) \odot \bar{B}_{\text{nov}}(z_s, c),$$

While our blending formulation references a continuum between novice and expert performance, it does not assume that skill lies on a single linear axis. We use multiple novice bases to capture diverse low-skill strategies (e.g., overcautious, inconsistent, or poorly timed behavior) and can employ multiple expert basis to represent distinct high-skill styles. Each subskill dimension modulates its own blend between these bases, enabling multi-dimensional, non-linear skill representations that remain interpretable and easily extensible to domains with multiple expert styles.

This blending formulation (shown in the dashed box in Fig 1) shifts the focus from reproducing every trajectory detail to capturing high-level, skill-relevant structure. Behavior prediction serves as the primary training signal, tying the embedding to stable, behaviorally meaningful variation across contexts while abstracting away noise. Crucially, this predictive design also yields a structured generative model of behavior: by intervening in the embedding space, we can probe how skill changes

would alter behavior, or hold skill fixed to forecast behavior in new contexts. These properties make the learned space useful for analysis, coaching, and simulation of counterfactual skill trajectories.

### 3.3 COUNTERFACTUAL TRAINING FOR SUBSKILL DISENTANGLEMENT

Human coaches diagnose deficits in specific subskills and tailor practice to address them (Ericsson et al., 1993; Newell, 1991; Wulf, 2016). More generally, effective assistance requires identifying which subcomponents of skill need improvement and designing targeted interventions (Anderson, 1982). This motivates our approach: to enable meaningful feedback, we model skill as a composition of subskills that can be selectively manipulated to predict changes in behavior.

Our goal is to learn a disentangled skill representation in which each subskill is captured by a dedicated slice of the latent space, such that manipulating that slice selectively modulates the corresponding behavior. Prior work on disentanglement, such as InfoGAN, $\beta$-VAE, and FactorVAE, has explored encouraging statistical independence between latent dimensions. While partially successful, these approaches face important drawbacks: they often limit the information capacity of the latent space Higgins et al. (2017), can be unstable to train (Locatello et al., 2019), and, crucially, do not naturally support counterfactual reasoning - that is, asking how a prediction should change if a single component of the latent space were modified Locatello et al. (2019). Even when disentanglement is encouraged, a second challenge arises: identifiability, or knowing which part of the latent space corresponds to which subskill. Conditional VAEs (Sohn et al., 2015) and related methods address this by supervising certain latent dimensions with labels (Kingma et al., 2014), anchoring them to known factors. However, this anchoring does not guarantee true disentanglement. Latent slices may still leak information about other factors, especially when labels are noisy or correlated, leading to entangled and ambiguous representations despite explicit supervision.

We propose a counterfactual training scheme motivated by Kim & Mnih (2018) that encourages both disentanglement and identifiability, while still optimizing for reconstruction accuracy. The embedding space is explicitly partitioned into subskill-specific slices,

$$z_s = \left[\, z_s^{(1)},\, z_s^{(2)},\, \ldots,\, z_s^{(K)} \,\right],$$

where each slice $z_s^{(k)} \in \mathbb{R}^{d_k}$ is intended to represent subskill $k$, and $\sum_k d_k = d$.

Reconstructed trajectories $\hat{\tau}_{z_s}$ are mapped through a predictor network $h_\psi$ to obtain subskill metrics $\hat{m}$ (Fig. 1), which serve as behaviorally grounded supervision signals during training. Each skill metric is defined in collaboration with domain experts and reflects a measurable behavioral quantity that serves as a proxy for an underlying subskill (e.g., steering reversal rate for control coordination, peak lateral acceleration for vehicle handling, or gaze dispersion for visual attention). These metrics provide weak yet semantically meaningful supervision that anchors each subskill dimension to interpretable aspects of human behavior.

To enforce counterfactual consistency, we perform subskill swaps between a randomly chosen pair of training examples: an *original* sample (the one being modified) and a *donor* sample (from which a single subskill slice is borrowed). For a selected subskill $k$, we replace the $k$-th slice of the original embedding with that of the donor:

$$\tilde{z}_{\mathrm{orig}}^{(k)} = z_{\mathrm{donor}}^{(k)}, \quad \tilde{z}_{\mathrm{orig}}^{(\ell)} = z_{\mathrm{orig}}^{(\ell)} \ \ \forall \ell \neq k,$$

and apply the same operation to the associated skill metrics to ensure supervision remains consistent:

$$\tilde{m}_{\mathrm{orig}}^{(k)} = m_{\mathrm{donor}}^{(k)}, \quad \tilde{m}_{\mathrm{orig}}^{(\ell)} = m_{\mathrm{orig}}^{(\ell)} \ \ \forall \ell \neq k.$$

In practice, we interleave counterfactual and standard training: with probability $p$, a batch is trained using the regular reconstruction and metric objectives, and with probability $(1-p)$, a batch is trained with counterfactual swaps (Alg. 1, Lines 7–8). This procedure creates counterfactual examples where the original participant retains their overall embedding but adopts one subskill dimension from the donor, allowing the model to learn how isolated subskills influence predicted behavior and corresponding metrics.

This balance ensures that the model maintains reconstruction fidelity while also learning to enforce subskill disentanglement. Since no ground-truth trajectory exists for this counterfactual, we do not apply a reconstruction loss to $\hat{\tau}_{z_s}$ for the swapped batch items (Alg. 1 Line 10). Instead, the

predictor network, $h_\psi$, is required to output the swapped metric for subskill $k$, forcing the model to adjust behavior in a way that matches the intervention.

Unlike approaches that impose structural constraints directly on the embedding space (e.g., linear independence penalties or orthogonality objectives), our method enforces disentanglement through behavior. The reconstructed trajectories must reproduce the correct subskill metrics, which forces each latent slice to encode and express its designated subskill in a behaviorally grounded way.

### 3.4 MODELING DETAILS AND LOSSES

The overall training objective combines several complementary losses that promote reconstruction fidelity, semantic alignment, and disentanglement:

$$\mathcal{L} = \lambda_{\text{traj}}\mathcal{L}_{\text{traj}} + \lambda_{\text{metric}}\mathcal{L}_{\text{metric}} + \lambda_{\text{MI}}\mathcal{L}_{\text{MI}}, \tag{1}$$

where $\mathcal{L}_{\text{traj}}$ is a trajectory reconstruction loss between predicted and observed behaviors, $\mathcal{L}_{\text{metric}}$ supervises subskill-specific metrics predicted by $h_\psi$, and $\mathcal{L}_{\text{MI}}$ is a mutual-information term encouraging consistency between the embedding $z_s$ and reconstructed behavior. During counterfactual training steps, $\mathcal{L}_{\text{traj}}$ is omitted since no ground-truth trajectory exists for the swapped embedding, and only the metric loss for the swapped subskill is applied.

Our model integrates participant-level skill embeddings with trajectory and context encoders built from established sequence architectures. Contextual map features are processed using a Point-Net–Transformer encoder (Gao et al., 2020; Gopinath et al., 2025), which captures both local geometry and global layout. The trajectories are predicted via two decoders, which produces elementwise blending weights for expert–novice interpolation.The skill metrics predictor uses an LSTM to reprocess generated trajectories and provide supervision. Architectural choices, training settings, and loss coefficients are detailed in Appendix A.2

## 4 DOMAINS AND DATASETS

### 4.1 HIGH-PERFORMANCE RACING

High-performance racing is an ideal domain for studying skill because it requires the integration of multiple subskills under demanding conditions. We focus on six core subskills identified by expert coaches and prior work (Schrum et al., 2025): (i) vehicle handling, (ii) gaze control, (iii) know-how (strategic knowledge of racing lines and techniques), (iv) control inputs (coordination of steering, throttle, and braking), (v) physical ability, and (vi) perceptual ability. These determine a driver's overall competence and provide a structured target for disentangled representation learning.

Each behavioral trajectory $\tau_i$ consists of vehicle pose, speed, and control signals downsampled to 100 points per track segment. The context $c$ for this dataset refers to the racetrack that the behavioral trajectory was executed on. We collected a dataset of racing behavior from 95 participants spanning novices to experts, using a high-fidelity driving simulator. Data collection proceeded in two phases: 70 participants each completed at least four laps on a single track modeled after a nearby raceway, and 25 participants each completed four laps across four distinct tracks at the same venue. This design provided both breadth (a large participant pool) and depth (multiple laps and multiple contexts), resulting in 1545 laps. The simulator provided realistic vehicle dynamics under repeatable conditions, enabling controlled yet ecologically valid measurement of driver behavior.

To connect observed behavior to underlying subskills, we leverage a set of behaviorally grounded skill metrics $m \in \mathcal{M}$, defined in collaboration with expert coaches (Schrum et al., 2025). Each metric is linked to a targeted task designed to probe a specific subskill: for example, peak lateral g-force in a skidpad drill reflects vehicle handling, steering reversal rate in a slalom drill reflects control input coordination, gaze fixation during driving sessions reflects gaze policy, occlusion task accuracy (where the visual scene was briefly hidden) reflects perceptual speed, dynamometer output reflects physical strength, and written test scores reflect strategic know-how of racing lines and techniques. These metrics (among others) provide partial, noisy evidence about latent subskills. Collectively, they supply the supervision signals necessary for learning structured representations of $z_s$.

Table 1: Results across desiderata in two domains, Racing (R) and Baseball (B). Higher is better for ↑, lower is better for ↓. Bold = best, underline = second-best.

| | SAIL (ours) | | SAIL w/o CF | | SAIL w/o basis | | SimCLR | | β-VAE | | VAE | | AE-LC | |
|---|---|---|---|---|---|---|---|---|---|---|---|---|---|---|
| | R | B | R | B | R | B | R | B | R | B | R | B | R | B |
| *Construct Validity* | | | | | | | | | | | | | | |
| Silhouette (↑) | 0.72 | | **.77** | | 0.67 | | 0.40 | | 0.74 | | 0.75 | | 0.67 | |
| Test–retest similarity (↑) | **.995** | **1.0** | 0.995 | 0.999 | 0.990 | 0.996 | 0.928 | 0.96 | 0.928 | 0.66 | 0.839 | 0.98 | 0.891 | 0.998 |
| **Composite (Construct)** (↑) | 1.86 | **1.0** | **2.0** | 0.997 | 1.69 | 0.99 | 0.57 | 0.88 | 1.49 | 0.00 | 0.95 | 0.94 | 1.06 | 0.99 |
| *Predictive Utility* | | | | | | | | | | | | | | |
| Behavior prediction (RMSE ↓) | **2.76** | 0.24 | 2.75 | **0.22** | 5.05 | 0.32 | 4.15 | 0.29 | 4.48 | 0.26 | 4.61 | 0.26 | 4.87 | 0.31 |
| OOC generalization (RMSE ↓) | **6.37** | 0.33 | 6.50 | **0.29** | 12.77 | 0.48 | 10.27 | 0.38 | 12.51 | 0.30 | 12.42 | 0.36 | 14.12 | 0.37 |
| **Composite (Predictive)** (↑) | **2.0** | 1.59 | 1.98 | **2.0** | 0.17 | 0.00 | 0.89 | 0.83 | 0.46 | 1.55 | 0.41 | 1.23 | 0.08 | 0.68 |
| *Disentanglement & Interpretability* | | | | | | | | | | | | | | |
| Alignment Ratio (AR ↑) | **3.25** | **2.17** | 1.24 | 1.58 | 1.11 | 1.43 | 1.73 | 0.94 | 1.12 | 0.59 | 1.05 | 1.04 | 2.40 | 1.79 |
| Targeted Change Index (TCI ↑) | **.93** | 0.13 | 0.89 | 0.14 | 0.73 | 0.15 | 0.45 | 0.16 | 0.63 | 0.19 | 0.78 | 0.19 | 0.73 | **.19** |
| Relative Influence Ratio (RIR ↑) | **2.11** | **.25** | 1.76 | 0.08 | 1.67 | 0.07 | 1.85 | 0.12 | 1.86 | 0.11 | 1.61 | 0.07 | 1.56 | 0.12 |
| **Composite (Disentangle)** (↑) | **3.0** | 2.0 | 1.37 | 0.85 | 0.81 | 0.87 | 0.83 | 1.0 | 0.95 | 1.22 | 0.78 | 1.29 | 1.20 | **2.04** |

## 4.2 BASEBALL HITTING

We applied **SAIL** to a supplemental dataset of baseball hitting collected from 13 players on a competitive adult team in a semi-professional league. While all participants were skilled and experienced players, they were not yet at the level of an expert benchmark and thus exhibited substantial variation in the execution of key subskills. One highly skilled participant was identified as an expert and used to define the canonical expert basis for blending, while the remaining players provided a diverse set of novice-to-intermediate trajectories. In total, 74 batting trials were recorded, spanning both pitching machine sessions and tee batting conditions. Whole-body kinematics of swing motions were captured using an optical motion capture system. In collaboration with a coach, we identified three core subskills and associated metrics of effective hitting: (i) the *kinematic chain*, or the sequential transfer of momentum across body segments; (ii) *pelvis pausing*, or the ability to momentarily stabilize the pelvis to build rotational power; and (iii) *thigh pausing*, or the controlled deceleration of the lead thigh to anchor lower-body mechanics. The contexts $c$ for this dataset are batting from a tee and batting against live pitches from a machine. Unlike the racing dataset, which incorporates a broad set of tasks probing multiple subskills, this dataset is narrower in scope and primarily intended as a secondary domain to test the generality of our approach.

To address the limited size of the dataset and capture broader variability, we generated synthetic participants by applying trajectory augmentations (time warping, noise injection, and scaling) to recorded swings. For players with both tee and regular trials, we estimated a global offset between conditions and used it to synthesize additional regular swings. This procedure produced artificial batting trials that preserved the underlying structure of advanced but non-expert motion while introducing diversity reflective of natural variations in skill.

To our knowledge, there are no existing datasets that capture multimodal behavioral signals, targeted drills, and coach-aligned annotations for baseball that are comparable in richness to our racing dataset. We therefore treat this baseball dataset as supplemental—smaller in scale, narrower in subskill coverage, and heavily augmented with synthetic trials—but nevertheless valuable for demonstrating that **SAIL** can extend beyond driving to a distinct motor domain.

## 5 RESULTS

We compare our method against several baselines and ablations to evaluate the contribution of each component:

**SimCLR (contrastive baseline).** A self-supervised representation learning method that uses contrastive losses to encourage invariance within a person. We adapt SimCLR to trajectory data to test whether a generic contrastive objective is sufficient for extracting skill-relevant embeddings Chen et al. (2020a).

**β-VAE (disentanglement baseline).** An extension of the VAE with a stronger KL regularization term that encourages factorized latents. We include β-VAE as a canonical disentanglement method, testing whether generic disentanglement pressure yields interpretable subskills Higgins et al. (2017).

**AE (autoencoder baseline).** A standard trajectory auto-encoder (Kingma & Welling, 2013). This provides a comparison to unsupervised compression methods that capture per-trial variability but are not explicitly designed to model persistent skill or subskill structure Hinton & Salakhutdinov (2006).

**AE with linear constraints (structured baseline).** A recent extension of the autoencoder framework that incorporates linear constraints into the latent space to encourage semantically meaningful and identifiable embeddings. We include this method to test whether such constraints help discover interpretable subskills in driving data and utilize the subskill metrics to create the linear constraints (Lin et al., 2020).

**Ablation: without counterfactual training.** This variant removes the counterfactual swap objective, training only with behavioral prediction via the expert-novice bases blending. This isolates the contribution of counterfactual training to disentanglement and interpretability.

**Ablation: without expert–novice basis and without counterfactual training.** Instead of decoding trajectories as a blend of expert and novice bases, this variant decodes directly from the skill embedding. We also ablate the counterfactual training in this variant. This tests whether the basis decomposition is necessary for isolating skill-relevant variation from noise and style.

For all baselines that operate at the trial level (Sim-CLR, AE, $\beta$-VAE, AE-LC), we extract embeddings per trajectory and then poll them across laps for each participant, yielding a participant-level embedding

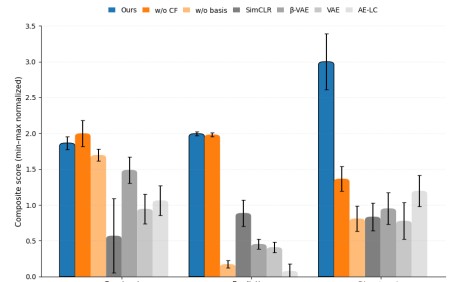

Figure 2: Composite scores across the three desiderata: construct validity, predictive utility, and disentanglement/interpretability. Bars show performance of our method (**SAIL**), ablations (w/o CF, w/o basis), and baselines (SimCLR, $\beta$-VAE, VAE, AE-LC). Higher is better for all desiderata.

comparable to our method. We evaluate our approach and baselines along the three desiderata introduced in Section 3: (1) construct validity, (2) predictive utility, and (3) disentanglement and interpretability. Together, these evaluations assess whether the learned representation $z_s$ is well-structured, useful for downstream tasks, and decomposable into meaningful subskills.

## 5.1 CONSTRUCT VALIDITY

We evaluate *construct validity* by measuring whether the learned embedding captures stable, skill-relevant structure rather than transient fluctuations or task-specific noise. Two complementary metrics are reported in Table 1. Full metric definitions of the metrics are provided in Appendix A.4.

In this work, we interpret construct validity in the behavioral and representational sense—whether the learned embedding behaves consistently with the theoretical construct of human skill (i.e., stable within individuals and discriminative across skill levels)—rather than as formal psychometric validation. Our goal is to provide empirical evidence that the learned latent space captures skill rather than transient performance fluctuations.

- **Silhouette score** (↑): clustering quality by skill group.
- **Test–retest similarity** (↑): stability of embeddings across repeated trials.

**Discussion:** As shown in Figures 2 and Table 1, **SAIL** achieves the strongest composite score for construct validity in both racing and baseball. The participant-level embedding ensures high test–retest stability (0.995 in racing, 1.0 in baseball), reflecting that $z_s$ captures persistent aspects of skill rather than trial-specific noise. The ablation without counterfactual training (**SAIL** w/o CF) performs comparably, which is expected since counterfactual swaps are designed to improve disentanglement rather than stability. In contrast, removing the novice–expert basis (**SAIL** w/o basis) reduces silhouette scores, likely because the embedding must capture trial-level variability instead of abstracting away noise. Baselines such as SimCLR cluster trials from the same participant but may emphasize stylistic consistency rather than stable skill, while autoencoder variants show

some separation but less interpretability. Overall, these results confirm that anchoring skill at the participant level and incorporating basis blending is key to achieving construct validity.

## 5.2 PREDICTIVE UTILITY

We next assess *predictive utility* by evaluating whether the learned skill embeddings support accurate trajectory forecasting within and across contexts.

- **In-context prediction (RMSE ↓):** trajectory accuracy within the same context.
- **Out-of-context prediction (RMSE ↓):** generalization to novel contexts.

**Discussion:** As shown in Figures 2 and Table 1, **SAIL** achieves the lowest error for both in-context and out-of-context prediction, confirming that the learned embeddings capture stable behavioral tendencies that generalize across settings. The ablation without counterfactual training (**SAIL** w/o CF) performs similarly on predictive metrics, consistent with the design of counterfactual swaps, which target disentanglement rather than raw forecasting. By contrast, removing the novice–expert basis (**SAIL** w/o basis) substantially degrades prediction, with errors nearly doubling in the racing domain. This highlights that basis blending is critical for abstracting away trial-level variability and anchoring $z_s$ in structured behavioral dimensions. Baselines show weaker generalization: autoencoder variants (VAE, $\beta$-VAE, AE-LC) capture per-trial variability but fail to transfer to new contexts. Together, these results demonstrate that predictive validity is best achieved by combining person-level embeddings with structured novice–expert bases. Notably, predictive utility is not significantly reduce by our counterfactual training scheme, suggesting that it may encourage disentanglement without greatly hurting prediction accuracy. AE-LC on the other hand performs the worst in terms of predictive utility.

## 5.3 DISENTANGLEMENT AND INTERPRETABILITY

Finally, we evaluate whether the representation decomposes into interpretable subcomponents that correspond to distinct subskills. Three complementary metrics are reported in Table 1:

- **Alignment Ratio (AR ↑):** measures how well each subskill slice $z_s^{(k)}$ predicts its intended metrics compared to non-target ones, indicating subskill–metric correspondence.
- **Targeted Change Index (TCI ↑):** quantifies the effect of counterfactual swaps by checking whether trajectory changes are concentrated in the targeted features, with higher values reflecting more selective control.
- **Relative Influence Ratio (RIR ↑):** compares the relative impact of subskills on overlapping behavioral outputs (e.g., control inputs vs. gaze), capturing whether slices exert influence in proportion to their intended role.

**Discussion:** Figures 2 and Table 1 show that **SAIL** consistently achieves the highest disentanglement scores in racing, while differences are smaller in baseball. This gap can be explained by the nature of the datasets. The baseball data is smaller in scale and includes substantial synthetic augmentation, which produces clearer, more linearly separable relationships between metrics and behavior. As a result, structured baselines such as AE-LC are able to capture some of these relationships without the need for counterfactual supervision, narrowing the gap. By contrast, the racing domain contains richer and more heterogeneous variability, making counterfactual swaps essential for learning interpretable subskill slices. Overall, these results suggest that counterfactual supervision is particularly valuable in complex, high-variance settings, whereas in simpler or synthetic domains, weaker baselines can exploit linear structure to partially mimic disentanglement.

## 6 LIMITATIONS

Our evaluation is limited by dataset scale and scope, particularly in the baseball domain where data are small and augmented. The method also depends on noisy, predefined subskill metrics, and assumes a smooth novice–expert continuum that may overlook abrupt shifts or alternative strategies. Finally, while counterfactual training improves interpretability, it does not guarantee fully disentangled subskills.

## REPRODUCIBILITY STATEMENT

We have taken several steps to ensure the reproducibility of our results. All datasets used in our experiments are described in detail in Section 4 and Section A.3. Model architectures, hyperparameters, and training configurations are reported in Section 3 and further detailed in Appendix A.2.

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

# A APPENDIX

## A.1 USE OF AI ASSISTANCE

Portions of the text in this paper were refined with the assistance of ChatGPT. The tool was used only to improve clarity and readability of the manuscript and to aid in finding related works; all ideas, experiments, and analyses are the authors' own.

## A.2 ARCHITECTURE AND TRAINING DETAILS

**Skill embeddings.** Each participant is associated with a persistent, learnable skill vector $z_s \in \mathbb{R}^d$, stored in an embedding table initialized from $\mathcal{N}(0, 0.1)$. For the racing domain, $d = 12$, partitioned into six subskill slices: vehicle handling (3), gaze (3), inputs (3), know-how (1), physical (1), and perception (1). For the baseball domain, $d = 7$, partitioned into kinematic chain (3), pause pelvis (2), and pause thigh (2).

**Map encoder.** For racing, we use a PointNet–Transformer encoder that processes local lane geometry and produces per-segment encodings of dimension 64. This corresponds to the map encoder in our formulation (Section 3). For baseball, no map input is used.

**Blending decoders $g_\phi$.** Behavior is represented as a blend of canonical expert and novice bases. The *alpha decoder* maps $(z_s, c)$ into elementwise interpolation weights $\alpha$ and predicts coefficients for novice PCA bases grouped by feature (e.g., position, steering, throttle, brake, speed).

**Skill predictor $q_\phi$.** To regularize the latent space, we predict $z_s$ from blending coefficients $\alpha$ using a 2-layer Transformer encoder (hidden size 128, 4 heads, dropout 0.1) with a [CLS] token.

**Trajectory-to-subskill predictor $h_{\text{spi}}$.** In addition to decoding metrics directly from $z_s$, we include a trajectory-to-subskill head $h_{\text{spi}}$ that maps reconstructed trajectories $\hat{\tau}$ into predicted subskill metrics. This auxiliary supervision ties subskill metrics to observable behavior, complementing the $z_s \mapsto m$ decoders. In practice, $h_{\text{spi}}$ is implemented as a two-layer MLP applied to flattened trajectory segments.

**Training.** We train using Adam with learning rate $2 \times 10^{-4}$, weight decay $10^{-5}$, batch size 256 (racing) or 5 (baseball), and a maximum of 2000 and 7000 epochs respectively. Losses include: (i) trajectory reconstruction ($\lambda = 0.05$–$0.1$), (ii) subskill metric decoding with $h_\psi$ ($\lambda = 4.0$), (iii) trajectory-to-subskill decoding with $h_{\text{spi}}$ ($\lambda = 4.0$), (iv) trial time prediction ($\lambda = 0.1$–$2.0$), and (v) counterfactual supervision applied on 20% of batches. Auxiliary penalties include contrastive regularization ($\lambda = 1$), VICReg ($\lambda = 0.005$), orthogonality ($\lambda = 0.001$), and adversarial consistency ($\lambda = 0.01$). Training was performed on NVIDIA RTX A6000 GPUs.

## A.3 RACING DOMAIN AND SUBSKILL METRICS

Following prior work in HPDE (Schrum et al., 2025), we model racing expertise as a composition of six subskills: *know-how*, *physical ability*, *gaze policy*, *vehicle handling*, *control inputs*, and *perception*. Each subskill corresponds to a dedicated slice $z_s^{(k)}$ of the overall skill vector $z_s$, and is supervised using behaviorally grounded metrics. Consistency is treated as a cross-cutting property across subskills rather than a separate dimension.

**Know-how.** Procedural and declarative knowledge of racing lines and techniques, assessed via a written test ($m_{\text{know}}$).

**Physical.** Motor ability and endurance, measured through grip strength and related assessments ($m_{\text{phys}}$).

**Gaze.** Visual attention strategies, measured via dispersion, dwell time, and fixation on apex cones ($m_{\text{gaze}}$).

**Vehicle handling.** Car control at the limit, measured through raceline deviation, lateral g-force, and skidpad/slalom performance ($m_{\text{vh}}$).

**Control inputs.** Coordination of steering, throttle, and braking, measured via steering reversal rate, throttle smoothness, and braking stability ($m_{\text{inputs}}$).

**Perception.** Prediction and interpretation of the environment, assessed via occlusion tasks with hidden track segments ($m_{\text{perc}}$).

Participants completed repeated lap blocks in a high-fidelity simulator, interleaved with drills (skid-pad, slalom, occlusion) and questionnaires. This design enables both between-subject discrimination (novices vs. experts) and within-subject stability, allowing $z_s$ to capture persistent skill while abstracting away transient performance fluctuations.

## A.4 EVALUATION METRICS

We provide details for the evaluation metrics reported in Section 5.1.

**Silhouette score** ($\uparrow$): Measures clustering quality of embeddings with respect to skill group. In the racing dataset, a subset of participants was labeled as experts or novices based on prior driving experience. Silhouette values compare within-group cohesion to between-group separation; higher values indicate that embeddings of the same skill group are tightly grouped and well-separated from others. Because analogous labels are not available for baseball, we omit this metric there.

A larger ratio reflects clearer separation between expert and novice embeddings in the latent space, providing an additional measure of construct validity.

**Test–retest similarity** ($\uparrow$): Assesses temporal stability of embeddings across repeated trials from the same participant. We compute cosine similarity between embeddings estimated from independent subsets of trajectories. High values indicate that the representation reflects persistent aspects of skill rather than trial-specific noise.

**In-context behavior prediction (RMSE $\downarrow$):** Assesses how well $z_s$ can be used to predict trajectories in the same context from which it was derived (e.g., skill inferred from laps on one racetrack and evaluated on the same track). Lower error indicates that the embedding captures fine-grained behavioral tendencies that persist within a given setting.

**Out-of-context (OOC) generalization error (RMSE $\downarrow$):** Evaluates predictive performance when applying $z_s$ to novel contexts (e.g., skill inferred from behavior on one racetrack and tested on a different track). Lower error reflects better transfer, showing that the embedding encodes stable skill structure that generalizes beyond training conditions.

**Alignment Ratio (AR $\uparrow$):** Computed by training shallow linear probes on each subskill slice $z_s^{(k)}$ to predict its corresponding behavioral metrics. AR measures how strongly the intended metric is predicted relative to non-target metrics. High AR values indicate that each slice encodes the correct factors with minimal cross-contamination.

**Targeted Change Index (TCI $\uparrow$):** Computed by performing counterfactual swaps of individual subskill slices and measuring changes in trajectory features. TCI quantifies the proportion of change concentrated in targeted features versus off-target leakage, with higher values indicating that slices selectively govern their intended behavioral dimensions.

**Relative Influence Ratio (RIR $\uparrow$):** Assesses the relative dominance of subskills on overlapping behavioral feature sets. For example, manipulating the control inputs subskill should strongly influence brake/throttle/steering traces, while gaze manipulations should more strongly affect vehicle position. High RIR values indicate that the representation disentangles subskills while also capturing their relative strengths in shaping shared outputs.

**Composite Scores**. For each desideratum, we min–max normalize each constituent metric across methods (direction-corrected so higher is better) and sum the normalized values. The composite therefore ranges from 0 to the number of metrics in that desideratum (e.g., 2 for Predictive; 3 for Disentangle). In baseball, we omit Silhouette (no group labels), so Construct uses only test–retest (range 0–1). See Table 1 for the underlying metric values.

## A.5 BASEBALL DOMAIN RESULTS

While high-performance racing is our primary evaluation domain, we also tested **SAIL** on a supplemental dataset of baseball hitting. This domain probes a different set of motor subskills and provides a test of cross-domain generalization. Because the dataset is smaller in scale and narrower in subskill

coverage, performance differences are less pronounced than in racing. Nevertheless, **SAIL** achieves the highest overall composite score, balancing predictive accuracy with disentanglement, whereas ablations highlight the trade-off between predictive utility (w/o CF) and interpretability.

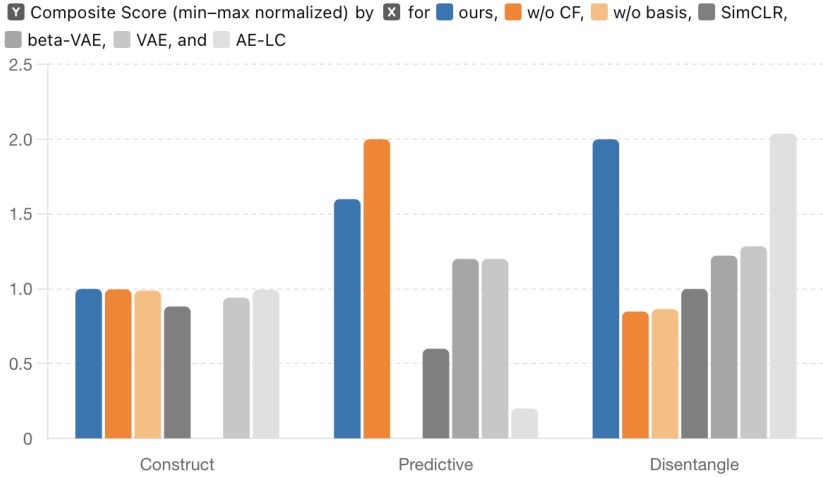

Figure 3: Composite scores across desiderata for the baseball domain. Bars show performance of our method (**SAIL**), ablations (w/o CF, w/o basis), and baselines (SimCLR, $\beta$-VAE, VAE, AE-LC). Although differences are smaller than in racing, **SAIL** maintains the best overall balance across desiderata.

