# OpenReview forum: "Disentangled Skill Representations for Predictive Human Modeling"
_ICLR.cc/2026/Conference — ICLR 2026 Conference Withdrawn Submission_

### Official Review · Reviewer_7LuY · 2025-10-31

**Soundness:** 2
**Presentation:** 2
**Contribution:** 3
**Rating:** 2
**Confidence:** 3

**Summary:**

The paper aims to introduce a method to produce skill embeddings that reflect *human* skill. These representations are designed to be interpretable, composable (learned in terms of preset sub-skills), persistent and individual to each participant, and have strong predictive capabilities. Each skill is learned as a blend of expert and novice behaviour bases and disentangled using counterfactuals. Two datasets were used, racing and baseball; the baseball dataset was augmented in a variety of ways.

**Strengths:**

- Good ablation studies.
- Experimental results seem promising, with a good number of baselines.
- The use of counterfactuals is an interesting idea.

**Weaknesses:**

- The method was difficult to understand since appropriate context isn't given at places. See below for more details.

- There are several typos that should be addressed. I have listed a few here:
	- Line 36: missing full stop.
	- Line 415: "Discussion:" repeated

- I believe that the following prior work are relevant to this paper and should be considered to be mentioned in Related Work (in order of relevance in my opinion):
	- [Petangoda, Janith C., et al. "Disentangled skill embeddings for reinforcement learning." _arXiv preprint arXiv:1906.09223_(2019).](https://arxiv.org/abs/1906.09223)
	- [Hausman, Karol, et al. "Learning an embedding space for transferable robot skills." _International Conference on Learning Representations_. 2018.](https://openreview.net/forum?id=rk07ZXZRb)
	- [Gupta, Abhishek, et al. "Meta-reinforcement learning of structured exploration strategies." _Advances in neural information processing systems_ 31 (2018).](https://proceedings.neurips.cc/paper/2018/hash/4de754248c196c85ee4fbdcee89179bd-Abstract.html)
	- [Dave, V., & Rueckert, E. (2025). Skill Disentanglement in Reproducing Kernel Hilbert Space. _Proceedings of the AAAI Conference on Artificial Intelligence_, _39_(15), 16153-16162.](https://ojs.aaai.org/index.php/AAAI/article/view/33774)

- Section 3, Problem Formulation:
	- Line 148: The math notation isn't written well. The notation $D$ and $c$ aren't used in the notation for $\mathcal{D}$ and $\tau_i$ even though the former are referred to when defining the latter.
	- Is $\mathcal{M}$ a space of functions? I.e., is a skill metric $m\in \mathcal{M}$ a function?
	- What is $k$ in $z_s^{(k)}$ denoting? How are the subcomponents composing to create a skill?
- Line 247: "... can be unstable to train ..." — can you please provide a reference here.
- The authors don't provide an algorithm that describes the end-to-end process of their method. Having one would help readability a lot. I was generally unable to decipher what overall method is. A lot of different components were described, however I was not able to understand how they all fit together; Figure 1 was not helpful since it is not referred to in the text in an explanatory matter, and it also contains notation that wasn't used in the main text.
- Figure 1 refers to some notation (e.g., $z_{donor})$ that isn't explained / defined in the text.
 - There is a lack of variance / standard error being reported. Were repeated experiments carried out?
 - There is a lack of motivation on what / how the learned skill representations can be used.
- The number of datasets used isn't sufficient, especially since the Baseball dataset is (self-admittedly) too small and required augmentation. Are there other datasets that could be evaluated on and added to the paper?

**Questions:**

- Line 36 - '... from other variables in representation learning...' — what are the 'other variables' referred to here? Could the authors state some examples here by extending the sentence.
- Line 40 - It is unclear what 'performance' means here. From reading ahead, this is a technical word from another paper, and it is not used in the usual way the word is used. I think at least adding a reference to this paper is necessary here to indicate this distinction. You could also paraphrase a short definition, or use a footnote to elaborate further.
- Line 81: "Traditional metrics such..." — traditional metrics of what?
- Line 152:  I am confused about the comparison of *skill* with *performance*. It isn't apparent why these are concepts that require a distinction made; they are obviously very different concepts (e.g., a skill is something a system has / can learn, and performance measures how good that skill is). Is there a context where performance
- Line 169: "... assigns each participant ..." — what is a participant here? Are the trajectories sampled from different individuals? If so, this fact should be reflected in the notation introduced in the Problem Formulation section.
	- Are there multiple $z_s$, one for each participant?
	- Is the number of participants known at training time + is the training data indexed by each participant?
- Line 183: Is this the lower bound? If so, please give it an Equation number and refer to it in the preceding sentence.
- Section 3.3:
	- How is each $d_k$ chosen?
- For the baseball dataset, what do the results look like with just the original dataset, without the augmentations?
- Line 476: "Finally, while counterfactual training improves interpretability, it does not guarantee fully disentangled subskills." — In Table 1, SAIL w/o CF had worse disentangle scores. Thus what is the reason for making this claim?

---

> ### Author Response · Authors · 2025-11-20
> **Response to Review 7Luy**
>
> We appreciate the reviewer's comments and have addressed each critique below:
>
> **Critique 1: An algorithm would improve clarity:** We appreciate the reviewer’s feedback and have added an algorithm overview (Algorithm 1) to the paper to clarify the training procedure. The new section summarizes the SAIL pipeline, including participant embedding initialization, expert–novice blending, subskill metric decoding, and the counterfactual update step. We believe this addition makes the training loop and its connection to the loss components clearer
>
> **Critique 2: Lack of variance reported**
> We appreciate the reviewer’s feedback and have added standard error bars to the plot.
>
> **Critique 3: Notation and clarity could be improved**
> We appreciate the comments from the reviewer and have updated the paper (in red) based upon these suggestions.
>
> **Critique 4: Difference between performance and skill needs to be clarified.**
> We appreciate this feedback and have added the following footnote to the paper with citations: “\footnote{In psychology and motor-learning theory, performance is considered a transient expression of skill influenced by situational factors \cite{iso2024theory,fitts1967human} We adopt this usage throughout.}”
>
> **Critique 5: Add citations on robot skill work**
> “Several works in reinforcement learning and robotics have explored learning latent skill spaces for control policies. Hausman et al. (2018) and Petangoda et al. (2019) learn disentangled or transferable skill embeddings to enable policy reuse and compositional action generation across tasks, while Gupta et al. (2018) develop meta-reinforcement learning strategies that encourage structured exploration. More recently, Dave and Rueckert (2025) propose a kernel-based approach for skill disentanglement within continuous control. However, these approaches address robotic control skill—the discovery of reusable action primitives or policies for task execution—rather than human skill as a persistent, compositional, and interpretable construct. Our work differs fundamentally in scope and objective: SAIL seeks to model how human skill manifests, evolves, and generalizes across contexts, focusing on cognitive and behavioral interpretability rather than motor control or policy transfer.”
>
> **All other comments related to notation, formatting, and typos have been addressed in the paper**

---

### Official Review · Reviewer_jCoF · 2025-11-01

**Soundness:** 2
**Presentation:** 2
**Contribution:** 2
**Rating:** 2
**Confidence:** 3

**Summary:**

The paper proposes SAIL (Skill Abstraction with Interpretable Latents), a framework for learning disentangled human skill representations from behavioral trajectories. Each person is modeled with a persistent skill embedding that blends expert and novice behavior bases to predict trajectories. To achieve interpretability, it uses counterfactual subskill swaps that encourage disentangled latent dimensions corresponding to specific subskills. The method is evaluated on two domains, including high-performance driving and baseball batting. Performance evaluation with baselines (SimCLR, β-VAE, and autoencoders) show that SAIL achieves superior stability, predictive utility, and interpretability.

**Strengths:**

- Clear formulation of “skill” as a representation problem: The authors explicitly define desiderata (construct validity, predictive utility, interpretability) and design the model accordingly.
- Innovative use of counterfactual swaps: The disentanglement approach is conceptually sound and provides interpretability beyond standard VAE or contrastive methods.
- Empirical demonstration across two distinct domains: Evaluating both driving and batting demonstrates cross-domain generality.

**Weaknesses:**

- Simplistic definition of expert and novice bases: The method defines the expert basis using a single ‘most expert’ participant and the novice basis via PCA over low-skill trajectories. This is a very simplistic and subjective setup, risking bias and poor generalization if expert behavior is heterogeneous.
- Weak comparative analysis: The baselines (SimCLR, β-VAE, AE-LC) are generic representation learning methods not optimized for skill abstraction. There is no comparison with recent hierarchical or meta-learning models for human skill, so the claim of superiority is not fully justified.
- Limited data and validation diversity: Only two domains, including synthetic simulated driving and small baseball batting, are tested. Also, real-world transfer or generalization to unseen participants is not well explored.
- Counterfactual procedure underexplained: The implementation of subskill swapping is intuitive but lacks quantitative justification, e.g., how sensitive results are to swap probability or latent dimensionality.

**Questions:**

- How robust is SAIL to the selection of expert and novice bases?
- Why were no skill-modeling or hierarchical imitation learning methods included for comparison?
- How are noisy or correlated subskill metrics handled to prevent interference across latent slices?

---

> ### Author Response · Authors · 2025-11-20
> **Response to jCoF**
>
> We appreciate the reviewer's comments and have addressed each critique below:
>
>
> **Critique 1: Simplistic definition of expert and novices bases**
> We thank the reviewer for this observation. Our goal was to introduce a transparent and interpretable instantiation of the expert–novice basis framework (single expert + PCA novices). However, more complex or learned basis definitions could be explored - for example, jointly learning expert and novice bases during training, deriving them via clustering or dictionary learning, or generating canonical behaviors from a model-based or optimal controller. However, our results show that even this simple setup performs strongly across both domains (Table 1), suggesting that the inductive bias of blending between interpretable endpoints is effective.
> Furthermore, in both domains, expert demonstrations exhibited strong behavioral convergence—experts tended to produce similar trajectories under the same context (e.g., in racing, following the same racing line and braking/throttle in similar zones) so a single expert basis effectively captured the canonical solution. In other domains in which expert solution may be heterogeneous, our approach allows for multiple expert basis.
>
> **Critique 2: No comparison with recent hierarchical or meta-learning models for human skill**
> We appreciate the reviewer’s point and agree that there exist hierarchical and meta-learning frameworks relevant to human skill modeling. However, these methods typically operate at a task-learning level rather than at the representation level targeted in this work. Our goal is to isolate and evaluate the representation of skill itself as a persistent construct, independent of task policy optimization or reward structure. For this reason, we compare against established representation learning baselines (SimCLR, β-VAE, AE-LC) that are widely used to assess disentanglement, stability, and predictive quality in latent spaces.
>
> **Critique 3: Limited data and validation diversity**
> We appreciate the reviewer’s point regarding data scale and diversity. As reviewer uaep noted, “This paper evaluates in two distinct domains, which is much better than most papers which only do one.” Both domains serve as out-of-distribution evaluation settings—all quantitative analyses are conducted on held-out participants not seen during training, ensuring that results reflect generalization to unseen individuals rather than overfitting. Moreover, our out-of-context (OOC) generalization experiments (Table 1) demonstrate the model’s ability to transfer to unseen track contexts, despite the limited data and domain coverage. These findings suggest that the learned representation captures underlying skill structure robustly even under distributional shifts. We agree that extending to additional domains and real-world driving data is an important next step, and we plan to pursue this in future work.
>
> **Critique 4: How are noisy or correlated subskill metrics handled to prevent interference across latent slices**
> Empirically, we find that even when some metrics are moderately correlated, the counterfactual swap objective isolates their effects behaviorally - each slice modulates its intended subskill features without destabilizing others.
>
> **Critique 5: How are noisy or correlated subskill metrics handled to prevent interference across latent slices**
> Empirically, we find that even when some metrics are moderately correlated, the counterfactual swap objective isolates their effects behaviorally - each slice modulates its intended subskill features without destabilizing others.

---

### Official Review · Reviewer_kGqf · 2025-11-01

**Soundness:** 2
**Presentation:** 2
**Contribution:** 3
**Rating:** 4
**Confidence:** 4

**Summary:**

This paper proposes a method to learn skill representations from a set of collected human behavior data. To encourage the skill representation to capture essential properties of skills rather than data noise, the authors propose several techniques to make the representation better align with human variations, such as variations across individuals and variations between novices and experts.

**Strengths:**

This work, to the best of my knowledge, proposes a novel way to learn representations for human skills. The individual-specific representation, the novice-expert interpolation, and the disentangled skill space provides useful techniques to model different aspects of the human skill space.

**Weaknesses:**

There are several weaknesses of the paper, mainly with respect to the disentanglement module, evaluation and presentation clarity.

1. Unlike traditional disentangled representation learning methods which typically are self-supervised, the proposed disentanglement module relies on human labeled subskill metrics.

2. The work claims that the learned skill representation is useful for downstream tasks, but there is no downstream task evaluations.

3. Regarding clarity, multiple places can be improved, including but not limited to
   1. Fig1 contains many undefied variables, such as $z_{donor}$ and $z_{orig}$.
   2. The training loss is completely deferred to the appendix, though it's an important component of the whole method.
   3. Several unclear metrics (see questions part for details).

Minor typos:

1. Table 1, in Behavior prediction (RMSE ↓) row, SAIL w/o CF (2.75) should be bold for Baseball  rather than SAIL (2.76).

2. Line 415, two "Discussion:"

**Questions:**

In Table 1, how are composite (Construct, Predictive, Disentangle) scores defined?

For the behavior prediction (RMSE ↓) metric which assesses how well $z_s$ can be used to predict trajectories in the same context from which it was derived. How is the prediction done?

In Table 1, for the baseball environment, why all methods has the same value for "Behavior prediction" and "OOC generalization"?

---

> ### Author Response · Authors · 2025-11-20
> **Response to kGqf**
>
> We appreciate the reviewer's comments and have addressed each critique below
>
>
>
> **Critique 1: The approach relies on human labeled metrics**
> We agree with the reviewer’s observation. Our approach does rely on behaviorally grounded metrics derived from domain experts. This design choice reflects the nature of the problem: human skill is a semantically defined construct, not an arbitrary latent factor. Incorporating these metrics anchors the learned dimensions to interpretable subskills (e.g., gaze control, vehicle handling), enabling counterfactual reasoning and coach-aligned explanations that purely self-supervised disentanglement methods cannot provide.  Furthermore,, these metrics are only required in the training set to ensure disentanglement and are not required at evaluation or deployment , where the model infers skill representations directly from behavior.
>
>
> **Critique 2: There are no downstream task evaluations**
> We appreciate the reviewer’s comment. Our goal is to evaluate whether the learned representation captures skill as a construct rather than to evaluate downstream performance on a specific task. Our evaluation already tests downstream usefulness indirectly through predictive behavior forecasting (Section 5.2) and counterfactual subskill manipulation (Section 5.3), which demonstrate that the learned skill embeddings can generalize across contexts and support targeted interventions. These are the core capabilities required for downstream applications such as adaptive coaching or personalized feedback. We agree that applying SAIL to concrete downstream tasks \ would further demonstrate practical value and we plan to include such evaluations in future work.
>
>
> **Critique 3: Clarity of the paper could be improved**
> We appreciate the reviewer’s feedback and have made each of these changes in the paper
>
>
>
>
> **Critique 4: In Table 1, how are composite (Construct, Predictive, Disentangle) scores defined**
> We appreciate the reviewer’s question and will add the following text to the appendix (A.4) to clarify: “\textbf{Composite Scores}. For each desideratum, we min–max normalize each constituent metric across methods (direction-corrected so higher is better) and sum the normalized values. The composite therefore ranges from 0 to the number of metrics in that desideratum (e.g., 2 for Predictive; 3 for Disentangle). In baseball, we omit Silhouette (no group labels), so Construct uses only test–retest (range 0–1). See Table \ref{tab:leaderboard-two-domains} for the underlying metric values. “
>
>
> **Critique 5: In Table 1, for the baseball environment, why do all methods have the same value for "Behavior prediction" and "OOC generalization"?**
> Thank you for the comment. This was a typo that we have corrected. It does not change the overall results.

---

### Official Review · Reviewer_uaep · 2025-11-04

**Soundness:** 2
**Presentation:** 3
**Contribution:** 2
**Rating:** 2
**Confidence:** 4

**Summary:**

This paper models human skill as "disentangled skill representations" and represents individuals with skill embeddings that are blends between experts and novices. It evaluates in driving and baseball batting.

**Strengths:**

* This paper evaluates in two distinct domains, which is much better than most papers which only do one.
* I believe human skill modeling is an important and interesting problem.
* The blending formulation seems novel.

**Weaknesses:**

* The paper's contributions are imprecisely stated to the point of being difficult to understand. For example, the point is repeatedly made that "our goal is to represent skill in a way that abstracts away trial-specific noise". However, this seems much less difficult than the paper seems to imply. For example, a simple average achieves this, or, more relevantly, an Elo score. I'm sure that doesn't satisfy other properties, but the work should more explicitly describe what the entire goal is.

* The idea to blend novices and experts is novel, but seems to be a rather large assumption. Wouldn't this imply a one-dimensional, or low-dimensional skill structure? And one in which skill behaves somewhat linearly? Is there empirical support to suggest that this is reasonable?

* The modeling and methodology sections are done well, but the results and evaluation are not up to the same standard. First, the "construct validity" section claims much more than it should (construct validity is difficult or impossible to measure in this way). Predictive utility, something that should be the main focus of the entire paper, is given a short paragraph. The results seemed rushed, which was unfortunate.

**Questions:**

What are the exact goals and contributions?
How limiting is the novices-experts blending? Could you blend more than just these two extremes?

---

> ### Author Response · Authors · 2025-11-20
> **Response to uaep**
>
> We appreciate the reviewer's comments and have addressed each critique below:
>
> **Critique 1: A simple average could abstract away trial level noise**
> We thank the reviewer for this suggestion. While simple averaging can reduce noise, it cannot distinguish between performance fluctuations that reflect risk-taking or exploration and those that reflect lack of control. For example, in high-performance driving, a spinout may increase average lap time even though it occurs because the driver is pushing the vehicle to the limit—a sign of advanced car control. Our model learns to contextualize such events, combining information across laps in a structured way rather than treating all deviations as errors. For instance, it can recognize that a spinout from an experienced driver who is pushing vehicle limits reflects exploratory, high-skill behavior rather than poor control—something a simple average would misinterpret as low skill.
>
>
> We appreciate the reviewer’s feedback that this is not clear in the text. We update the fourth paragraph of the Introduction to say: “While simple aggregation methods (e.g., averages or Elo scores) can smooth variability, they conflate skillful risk-taking with poor performance. For example, a spinout may raise average lap time even though it reflects an expert pushing the vehicle to its limits. Our approach learns a structured skill representation that contextualizes such events—abstracting away trial-level noise while preserving the subskill structure necessary to explain and predict behavior.”
>
>
>
>
> **Critique 2: The novice-expert basis implies low-dimensional skill structure**
> We appreciate the reviewer’s comment. Our blending formulation does not assume that skill is one-dimensional or globally linear. Rather, it provides a structured inductive bias where each subskill slice of the embedding independently modulates a blend between multiple novice and expert bases. The novice bases capture diverse low-skill strategies, reflecting that there are many ways to be a novice (e.g., overcautious, inconsistent, poorly timed). This allows the model to represent a rich, non-linear manifold of behaviors between the novice modes and the expert solution. In our current domains a single expert base suffices because expert strategies tend to converge. However, the approach naturally extends to multiple expert bases if the domain exhibits distinct high-skill styles.
>
>
> We will update Section 3.2 to state: “While our blending formulation references a continuum between novice and expert performance, it does not assume that skill lies on a single linear axis. We use multiple novice bases to capture diverse low-skill strategies (e.g., overcautious, inconsistent, or poorly timed behavior) and can employ multiple expert basis to represent distinct high-skill styles.  Each subskill dimension modulates its own blend between these bases, enabling multi-dimensional, non-linear skill representations that remain interpretable and easily extensible to domains with multiple expert styles.”
>
>
>
>
> **Critique 3: Construct validity section claims more than it should and predictive utility should be the focus**
> We appreciate the reviewer’s comment. We agree that predictive utility is an important criteria. However, predictive accuracy alone can be achieved even with entangled or misaligned embeddings (e.g., overfitting to context-specific cues). Our notion of construct validity therefore complements predictive utility by assessing whether the representation varies across individuals in theoretically consistent ways while remaining stable within a person across sessions and contexts. We do not claim psychometric proof of validity, but rather use behavioral indicators (e.g., test–retest reliability, expert–novice separation) as empirical evidence that the learned embedding reflects persistent skill rather than transient performance noise.
>
> We updated Section 5.1 to say: “In this work, we interpret construct validity in the behavioral and representational sense—whether the learned embedding behaves consistently with the theoretical construct of human skill (i.e., stable within individuals and discriminative across skill levels)—rather than as formal psychometric validation. Our goal is to provide empirical evidence that the learned latent space captures skill rather than transient performance fluctuations.”

---

### Note · Authors · 2026-01-07

I have read and agree with the venue's withdrawal policy on behalf of myself and my co-authors.